# Automated Technique for Identification of Prominent Nearshore Sandbars

**Nicole Zuck, Laura Kerr and Jon Miller \***

Department of Civil, Environmental and Ocean Engineering, Stevens Institute of Technology, Hoboken, NJ 07030, USA; nzuck@stevens.edu (N.Z.); lkerr@stevens.edu (L.K.)
\* Correspondence: jmiller@stevens.edu

**Abstract:** Nearshore sandbars are common features along sandy coasts. However, identifying sandbars within a beach profile traditionally requires a large historical dataset or subjective input from an observer. Several existing methodologies rely on reference profiles, which is problematic for new study sites with limited data sets and for nourished beaches that have drastic fluctuations in the cross-shore. This novel technique is suitable for beaches where a reference profile does not exist, as it identifies morphological sandbar features by a quantitative automated process. The technique identifies sandbars with a minimum steepness of 2% grade and a minimum height of 0.2 m. The morphological boundaries of sandbars were previously not well-defined, especially the seaward limit of the sandbar, contributing to difficulty in comparing surveys and sandbar morphologies. This technique standardizes the definitions of the bar limits mathematically via standard MATLAB functions, thus removing subjectivity and allowing results to be replicated. Bar identification is focused on the beach profile below the mean high water line, not cross on-shore positions, making the technique appropriate for nourished shorelines as well as those with large seasonal fluctuations. The automated technique was tested on 840 profiles collected near a recently completed beach nourishment project in Long Branch, NJ, USA. Results indicate success in identifying prominent sandbars within the test data set.

**Keywords:** sandbars; nearshore; MATLAB; coastal geomorphology; sandbar migration; beach state; bar state classification; beach survey; shoreline morphodynamics





## 1. Introduction

Nearshore sandbars are a common morphological feature of sandy coasts. They form within the active zone along beaches and play a vital role in beach dynamics and sediment exchange, especially as beaches respond to varying wave conditions and seasonal wave climates. Despite their prevalence along coastlines worldwide, the majority of research has focused on defining the bar crest and trough, with a surprising lack of focus on defining the seaward limit (herein referred to as the heel) of the bar. Here a method is proposed for automating the detection of sandbars and the extraction of their morphological characteristics. The novelty of this approach is that it can be used on beaches where a reference profile does not exist, and feature identification is a purely quantitative process. The selection of points to determine the extent of a sandbar is standardized, allowing for direct comparisons between surveys without subjective input from an observer. The automated technique accomplishes sandbar identification without the need for multi-year datasets or reference profiles, which are frequently not available. This makes the technique ideally suited for application in areas that have not previously been studied or in locations where significant changes have occurred (storm erosion, beach nourishment, etc.).

Previous studies determine the presence of a bar by first determining the presence of a longshore trough [1] or by comparing a measured profile to a reference profile and assessing the perturbations relative to the reference profile [2–6]. The studies that utilize a

reference profile define the seaward limit of the sandbar by identifying the point where the measured profile intersects the reference, but do not offer a quantitative solution for sites lacking a reference profile. Reference profiles are also not appropriate for use on nourished beaches due to the extreme changes in shoreline position and the location of the bars in the cross-shore.

One of the main challenges in defining the heel of a sandbar is that, unlike the crest and trough, the transition from bar to profile at the heel exhibits subtle changes in curvature that are difficult to detect. This technique provides an automated method to rapidly identify prominent sandbars and suggests a quantitative method of determining the heel. This methodology does not require manual selection by an observer to determine the heel location. In a multi-barred system, this technique assesses the innermost sandbar. For the purposes of this study, a prominent sandbar is defined as having a positive slope steeper than 2% grade between the trough and the crest and a minimum bar height of 0.2 m (Figure 1). This height threshold is based on the estimated measurement uncertainty and minimum height that could be confidently resolved by [2]. This prominent sandbar morphology corresponds to the intermediate states of [7], specifically State B: Longshore Bar-Trough and State C: Rhythmic Bar and Beach. Less prominent bars and flat bars (States D and E) may be identified by this methodology if they meet the height and slope thresholds. Characteristics of the classic beach states [7] are summarized in Table 1.

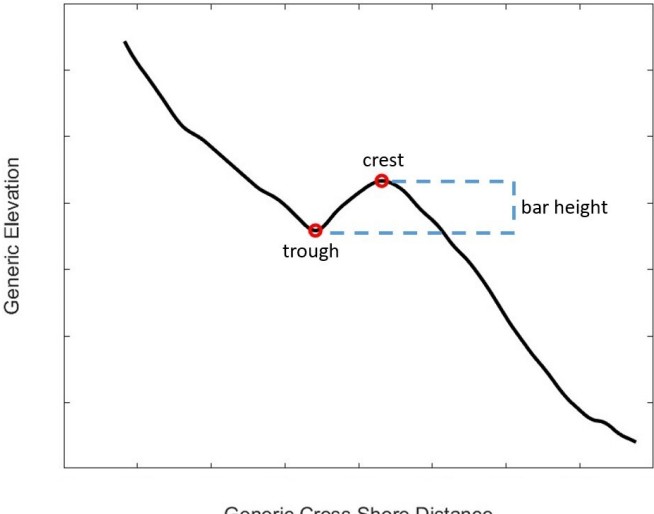

**Figure 1.** Sandbar height is defined as the vertical distance between the bottom of the sandbar trough and the sandbar crest. The trough is the local minimum in elevation in the trough, and the crest is the local maximum of the sandbar.

**Table 1.** Summary of beach states and general characteristics.

| Beach Category | General Description | Breaker Behavior and Surf Scaling Parameter ($\epsilon$) | Prominent Sandbar Present |
|---|---|---|---|
| A | Dissipative end member. Beaches exhibit very low gradients and may have multi-bar surf zones. Longshore variation is uncommon. | Spilling breakers. $\epsilon > 20$ | Unlikely |
| B | Longshore Bar-Trough. Beaches are characterized by steep beach faces and deep troughs. Cusps are common in the swash zone. There is little longshore variation. | Plunging breakers over the bar typically reform and become surging breakers on the beach face. $\epsilon \sim 2$ | Yes |

**Table 1.** *Cont.*

| Beach Category | General Description | Breaker Behavior and Surf Scaling Parameter ($\epsilon$) | Prominent Sandbar Present |
|---|---|---|---|
| C | Rhythmic Bar and Beach. Beaches are characterized by crescentic bars displaying longshore variation and deep troughs. The undulation of the bar often correspond to large cusps in the swash zone and along the beach face. | Breaker pattern varies with bar morphology. $\epsilon > 2.5$ | Yes |
| D | Transverse Bar and Rip. Beaches are characterized by mega cusps and strong rip currents. Bars display longshore variability and may weld onto the beach face between rip cells. | Breaker type varies with bar morphology. $\epsilon > 2.5$ | Likely |
| E | Ridge-Runnel or Low Tide Terrace. Beaches generally have a low berm and flatten below the low tide line, exhibiting a combination of low tide terraces and flat bars with corresponding runnels. | Plunging breakers. $\epsilon > 2.5$ | Unlikely |
| F | Reflective end member. Beaches exhibit steep beach faces and low gradient nearshore profiles. Beach cusps are common in the swash zone. A runnel is generally formed behind the sharp berm crest. | Surging breakers. $\epsilon < 2$ | No |

Identification of sandbars and assigning beach state categories have broader applications beyond monitoring bar evolution. For example, derived bar characteristics may aid in modeling wave interactions, specifically in assessing storm protection or changes observed on the shoreface [8]. Identifying areas of wave breaking on the seaward side of a sandbar further has implications for understanding the dynamics of the nearshore zone [3,8]. Quantifying the material storage within the sandbars of nourished beaches aids in identifying how much material is retained within the system, even if not located directly on the berm, and thereby the effectiveness of the nourishment. Prior studies also indicate the importance of incorporating the characteristics of the bar system into a shoreface nourishment design [9].

## 2. Data Collection and Initial Processing

Topographic and bathymetric data was collected at Long Branch, NJ, USA (Figure 2). The study area is approximately 4800 m in length, spanning from the landward edge of the berm to an approximate depth between −7.6 m to −10.7 m NAVD88. The constructed elevation of the nourished beach is 2.83 m NAVD88. The data set consists of 12 surveys collected between June 2020 and August 2021, each containing 70 standard profile transects (Figure 3).

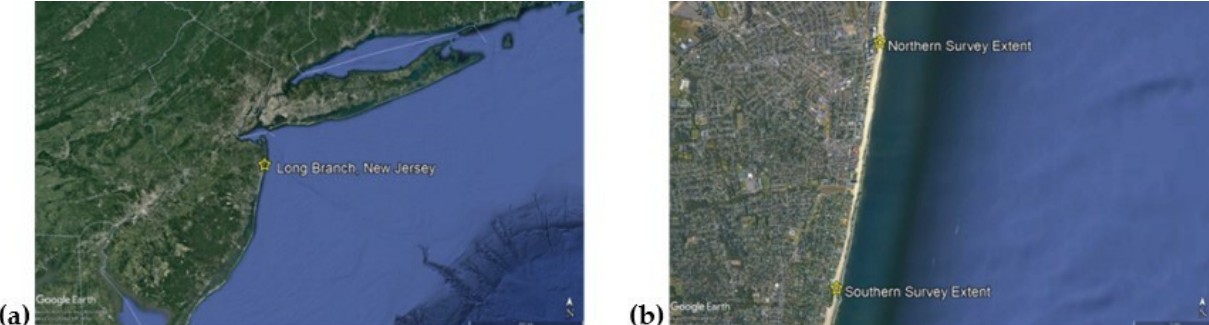

**Figure 2.** (**a**) The test data was collected at Long Branch, New Jersey. (**b**) The survey area spans approximately 4.8 km of beachfront, from the landward edge of the berm seaward to the approximate depth of closure.

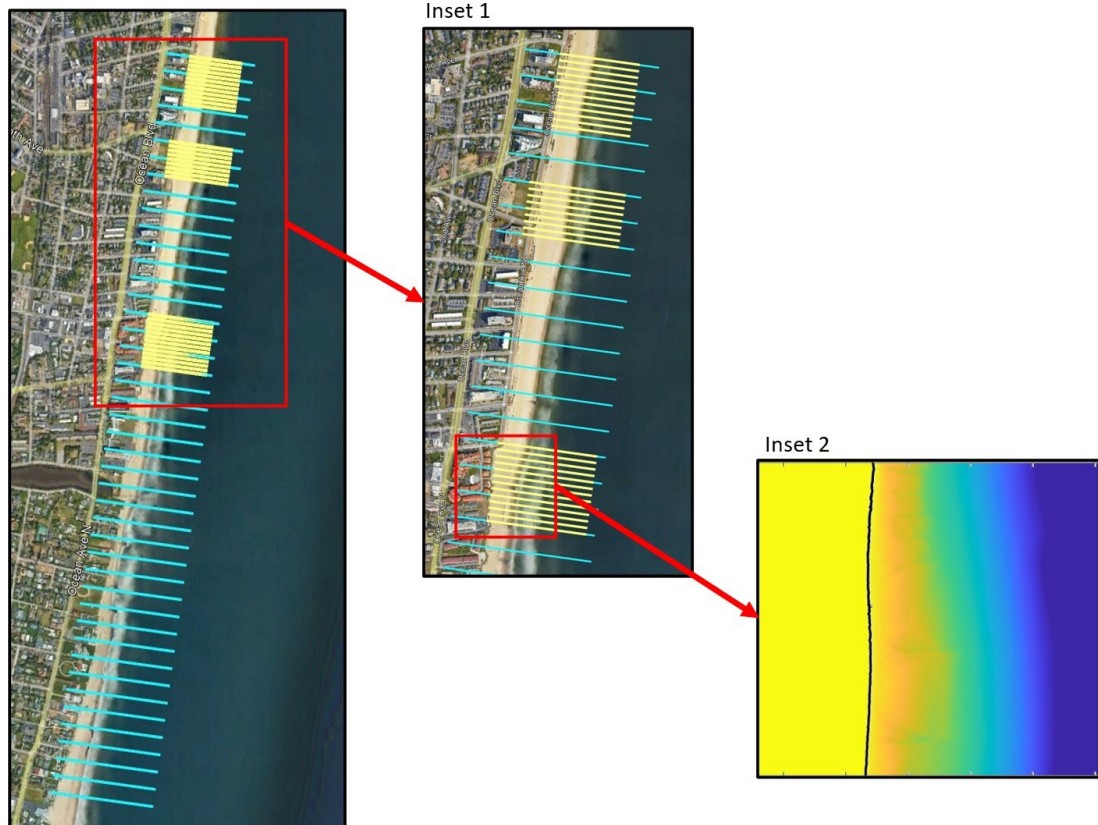

**Figure 3.** Planned survey transects at Long Branch, New Jersey. There are 70 total transects, including three high-resolution sections (first image inset). The line spacing is 91.4 m between standard lines (cyan) and 22.9 m between the high-resolution lines (yellow). The standard transects reach depths of approximately −10.7 m NAVD88 and the high-resolution transects reach approximately −7.6 m NAVD88 depth. The second inset image displays the DEM beach approximately one week post nourishment between the Line 26 and Line 40 transects. The black line indicates the approximate mean high water line (0.58 m NAVD88). The constructed berm elevation was 2.83 m.

The data was collected in three parts by the Stevens Coastal Engineering Research Group in accordance with their DUCKS System [10]: a Phantom 4 Pro uncrewed aerial vehicle (UAV or drone) equipped with a camera is used to survey the subaerial beach, a personal watercraft specially equipped for bathymetric data collection is used to survey the submerged profile, and a backpack mounted RTK GPS is used to survey the intertidal zone. Zimmerman et al. [11] summarizes the relative point densities and costs of these

survey methods. The UAV and backpack mounted RTK GPS achieve accuracy of 0.03 m and 0.05 m, respectfully [11]. The accuracy of the bathymetric measurements is 0.01 m [10].

All the points collected were combined to generate a digital elevation model (DEM) to analyze and capture the morphological features of the sandbars. The resolution of the DEM is 0.3 m-by−0.3 m. The profiles used in this study were extracted from the DEM along the planned survey transects. The profile data is saved as X, Y, and Z coordinates by line number. Sandbar features and the longshore and cross-shore position of the bar are extracted from the profile coordinates. At this step, it is recommended to identify the portion of the profiles that may contain a sand bar. This may be done by either masking sections on either end of the profile, or by extracting points and creating new X, Y, and Z variables. For the Long Branch data set, this was accomplished by applying a search window to identify points that were at both an elevation lower than the mean high water (MHW) datum (0.58 m NAVD88) and at a higher elevation than known rock debris in the area (−7.6 m NAVD88), thus only the central portion of the profile will be assessed to determine presence of a sand bar (Figure 4).

The upper limit is set at the MHW line to ensure that all potential sandbars are identified, including shallow bars visually observed during winter 2020–2021 surveys. Patches of rock debris are scattered throughout the study area; the depth limitation corresponds to the elevation of the shallowest rock structures. This depth limitation was applied for consistency between profiles and to avoid potential misidentification of the rock structures as sandbars. Other water level datums or depth limitations may be applied as appropriate and should be considered on a site-to-site basis.

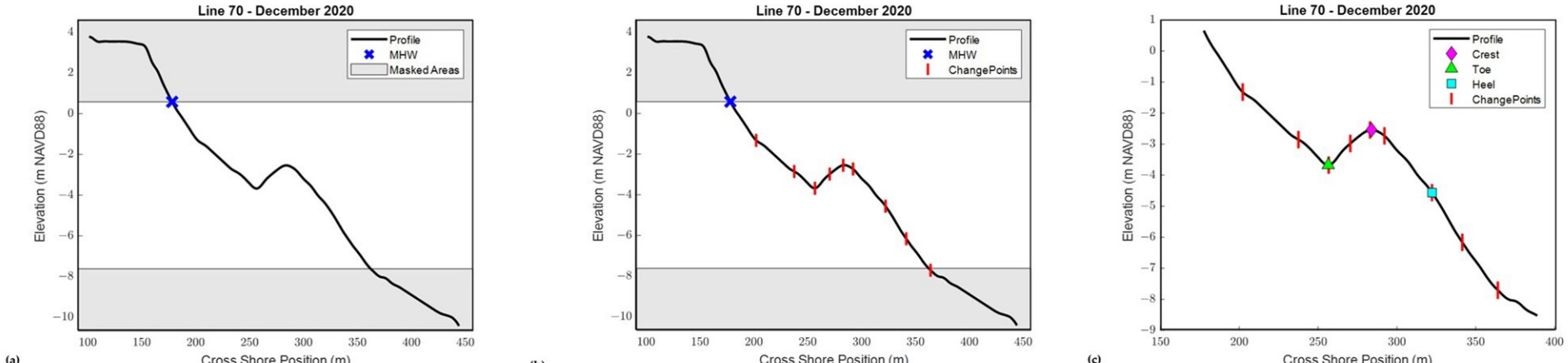

**Figure 4.** Example from the December 2020 survey displaying the sandbar identification process. (**a**) A search window is selected to determine the portion of the profile to be assessed. The upper shaded area represents the portion of the profile located above MHW (blue X). The lower shaded area corresponds to the maximum depth of known rock debris in the survey area. This lower portion is masked to limit potential misidentification of debris as sandbars. The lower masking is not required at all survey locations and should be considered on a site-to-site basis. (**b**) Nine changepoints (red vertical lines) were identified in the profile; masked (shaded) areas are not considered in bar identification. (**c**) Displays the final crest, toe, and heel locations. The crest (pink diamond) is a local maximum, the toe (green triangle) is the local minimum of the bar trough, and the seaward limit or heel (cyan square) is located at the first changepoint (red vertical lines) seaward of the crest and at a deeper elevation than the toe.

### 3. Sandbar Detection and Identification of Morphological Features

After the initial processing is complete and the desired data range is determined, each individual profile is assessed for the potential presence of a sandbar. The process is summarized in Figure 5. The profile data are entered into the *findchangepts* function in MATLAB, available in versions 2016A and later [12], to identify the changepoints within the profile (Figure 4b). A changepoint is defined as an abrupt change in signal, similar to curvature or the second derivative of the profile curve, however only the points of most significant change in the mean are returned [12] thus facilitating the process of sandbar identification.

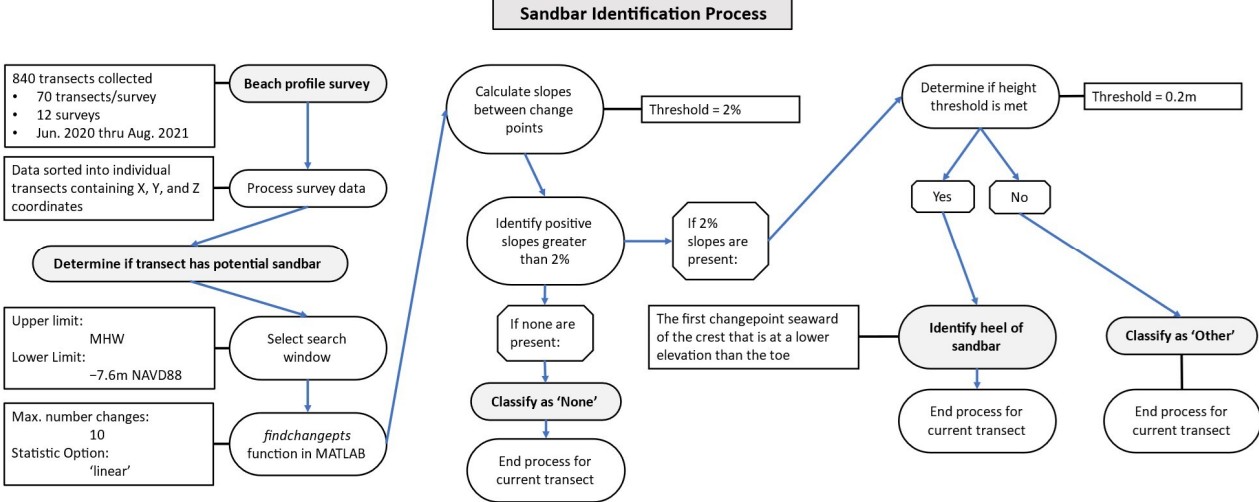

**Figure 5.** Flowchart summarizing the sandbar identification process.

The 'linear' statistic option is used to identify locations where the mean value of the points as well as the slope of the profile change most abruptly. The number of changepoints is limited to a maximum of 10. Iterative testing indicated that most profiles return less than 10 changepoints, and that when greater than 10 changepoints were present they did not have an impact on sandbar identification. The output of the function identifies the location of the changepoints in the input array. For example, the standard code reads (1):

$$CP\_locations = findchangepts([Depth\_Data],'MaxNumChanges',10,'Statistic','linear'); \quad (1)$$

thus, *CP_locations* can be used to extract the coordinates of the changepoints from the X, Y, and Z data. After identifying the changepoints, the slope of each line segment between each set of changepoints is calculated. Positive slopes greater than 2% grade indicate a potential sandbar. The slope threshold was selected for practical purposes. Iterative testing showed this threshold to be the lowest reliable value that was valid for identification of prominent bars Long Branch. This slope threshold is also perceptible during survey activities whereas shallower slopes are negligible to walkers. This allows for collaboration between field observations and measured data.

If a positive slope greater than 2% grade is identified (or if more than one is identified, the most landward slope is selected), the crest of the potential sandbar is then identified. The crest is the local maximum of the profile seaward of the positive slope and is identified with the *findpeaks* function (Figure 4c). A search window seaward of the positive slope is utilized to limit the number of peaks identified, and the highest peak is selected as the crest. Similarly, the landward extent of the potential sandbar (herein referred to as the toe) is a local minimum corresponding to the deepest part of the longshore trough immediately landward of the crest. The search window to identify the toe spans from MHW to the bar crest. The point of minimum elevation within this region is selected as the toe (Figure 4c).

After selecting the crest and toe, the bar height is calculated. The bar height is the vertical distance between the toe and the crest of the sandbar (Figure 1). If the height is greater than 0.2 m [2], the potential sandbar is categorized as 'True', and the process continues to identify the seaward extent (heel) of the sandbar. The bar height threshold is based on the estimated measurement uncertainty and minimum height that could be confidently resolved by [2]. If the height threshold is not met, the bar is categorized as 'Other', and the process ends for the current profile.

For sandbars categorized as 'True', the final step is identifying the heel of the bar. The heel is defined as the first changepoint seaward of the crest that is at a lower elevation than the toe (Figure 4c). Due to the extreme variance in sandbar morphologies, testing indicated that this definition is the most consistent and reliable throughout the dataset. After identifying the heel of the sandbar other sandbar geometric features may be calculated, such as bar width (horizontal distance from toe to heel) and volume. Identification of the sandbar heel is necessary to assess these bar features, as well as other morphological characteristics such as the bar shape parameter. The bar shape parameter is the ratio defined as the ratio between the bar width ($W_b$) and width from the bar crest to the heel ($W_{bs}$) (Figure 6) [3]. When the bar shape parameter ($W_{bs}/W_b$) is less than 0.5, the bar shape is skewed shoreward. When the ratio is equal to 0.5, the bar is symmetrical.

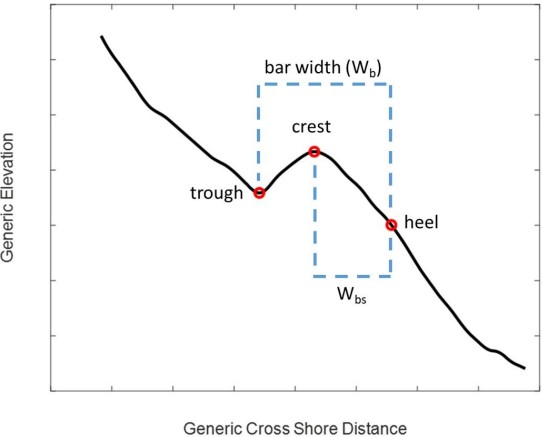

**Figure 6.** Sandbar width ($W_b$) is defined as the horizontal distance between the sandbar crest and the heel. The ratio between the bar width and the horizontal distance between the crest and the heel ($W_{bs}$) is used to calculate the bar shape parameter ($W_b/W_{bs}$).

## 4. Results and Discussion

The automated sandbar identification methodology was tested on 70 profiles extracted from survey DEMs created for each of the 12 surveys that were conducted between June 2020 and August 2021 at Long Branch, New Jersey (a total of 840 profiles). An observer review of the final plots of the profile with feature locations indicated the success of the automated process. The Long Branch beachfront was nourished during July and August of 2020, and the shoreline position changed drastically month-to-month. Therefore, a reference profile could not be used to identify sandbar features (Figure 7). This automated technique does not require historical data to generate reference profiles. Therefore, it may be employed for any study area, including nourished beaches such as our survey area. This technique relies on elevation datums, not cross-shore distances, to determine the search window and is appropriate for all stages of beach evolution following a nourishment project. The volume of data further stresses the need for automation; it is not only practical but also removes subjectivity from user input and provides consistency and repeatability in the selection of feature locations.

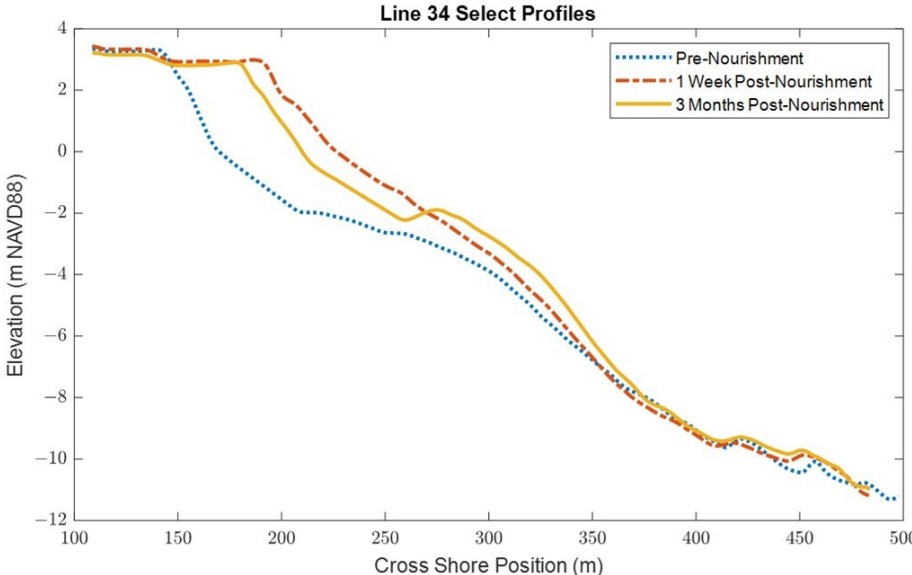

**Figure 7.** Profiles collected from within the nourishment area displaying the cross-shore variability due to nourishments at various phases of beach evolution: Pre-Nourishment (blue dotted line), 1 Week Post Nourishment (red dashed line), and 3 Months Post-Nourishment (yellow line).

The overall intent of this analysis is to identify large-scale bar features, namely the locations of the bar crest, toe, and heel. While some micro-level features may be indicative of the beach state, such as mega ripples in State C [7], this method is not optimized to detect micro-level features. The method is designed to assess large-scale bar behavior over considerable distances; therefore, the resolution of this method cannot confidently resolve micro-level features.

The metrics used to define the crest and toe of the sandbar are consistent with prior works [2–4]. A variety of techniques were tested to identify the heel position of the sandbar, and identification via changepoints was determined to be the most practical approximation across the varying sandbar morphologies within our dataset, which correspond to the intermediate beach states defined by Wright and Short [7]. The specific point that an observer may select as the heel is not necessarily the point selected by this technique, but it must be noted that an observer selection is subjective, and such a check process limits efficiency. This automated technique provides a close approximation of the heel position that is empirical and replicable based mathematically on the changepoint definitions. This methodology allows for direct comparisons between survey lines as well as between surveys and removes any bias of the user. A summary of other methodologies tested and rejected is provided in Table 2.

**Table 2.** Summary of additional methodologies tested to determine the heel position of a sandbar.

| Attempted Method | Process Description | Reason Abandoned |
| --- | --- | --- |
| Curvature (Second Derivative) | The curvature method was originally designed for selecting the toe and heel positions of sand dunes [13,14]. The point of maximum curvature is located where the steeper slope of the feature of interest intersects the flatter profile. | The slope of a sandbar often does not greatly vary from that of the profile; therefore, the point of maximum curvature is unclear or might be at a location in the profile unrelated to the sandbar. This method also requires an observer check. |
| Equal Elevation | This method takes the elevation of the toe and identifies the point on the profile seaward of the crest that is at the same elevation. | This method is only appropriate for highly symmetrical sandbars and ignores the overall negative slope of the profile. |
| Manual Selection | This method relies solely on observer input to visually identify and select the position of the heel. | This method is subjective and time consuming, and the results cannot be easily replicated. |

**Table 2.** *Cont.*

| Attempted Method | Process Description | Reason Abandoned |
|---|---|---|
| Extrapolation | The slope immediately landward of the toe is calculated, and a line is extrapolated from that slope. The line is set to intersect the toe. The length of the search window to determine this slope is equal to half the horizontal distance between the toe and the crest. | This method is only suitable for extremely peaky sandbars, and the extrapolated line does not always intersect the profile. Different search window lengths for the slope were also tested. Non-linear slopes were also tested to extrapolate the profile but were unsuccessful. |
| Equilibrium Beach Profile | This method compares the measured profile to an equilibrium beach profile (EBP), similar to the use of a reference profile to identify sandbars from the perturbations above the line. The EBP was calculated using local grain sizes and wave data. | The EPB does not represent the beach or nearshore environment. The measured profiles were much steeper than the EBP, and the EBP did not intersect the measured profile to identify perturbations or measure features. |
| Standard Distance | This method selects the heel of the sandbar based on a standard horizontal distance measured seaward from the crest location. The elevation of the profile matching the distance is selected for the heel point. Distances tested include $1\times$, $1.5\times$, and $2\times$ the horizontal distance between the toe and the crest. | This method is simple and efficient, but the accuracy is low, and it does not consider differences between sandbar morphologies. The width of sandbars is not a consistent value, nor is the symmetry. |
| Average Slope | This method is similar to Extrapolation; however, it uses the whole submerged profile rather than just a section landward of the toe. The average slope of the profile, calculated as the mean of the slope segments between changepoints, is used as the slope of a line intersecting the toe; all slope segments are included. The intersection of the line and the profile denotes the heel of the sandbar. | This method is more successful than the Extrapolation method but fails when assessing more shallow sloping or wide sandbars. Additionally, some profiles do not intersect with the line of average slope. Variations of this method included limiting the mean to only negative sloping segments and taking the average slope of the whole measured profile. They were also tested but proved unsuccessful. |
| Average Slope with Changepoint | This method is the same as the Average Slope method. However, in cases where there is no intersection, the furthest seaward changepoint is selected as the heel. | This method allows for the assessment of less peaky sandbars. However, the most seaward changepoint is not necessarily the best approximation of the heel. Due to the variation between profiles, selecting another $n^{th}$ position changepoint is not always possible, or would require an observer check to select the proper point, therefore introducing subjectivity and reducing efficiency. |

Identification of sandbars, including identifying the boundaries of the sandbar, is the basis for further assessment of a study site in understanding sediment transport, interactions with wave forcing, material storage, and other processes. Existing data-driven modeling techniques are capable of modeling large-scale morphodynamic changes [5,15] but come at greater time and computation costs. Additional sandbar characteristics that may be derived from the sandbar definition and identification proposed here include sandbar width, symmetry as bar shape parameter [3], and cross-shore position. From these characteristics, specifically information inferred from the bar shape parameter, migration patterns can be determined [16]. These derived characteristics may aid in a better understanding of the nearshore processes at a study area and can aid in determining beach categories [1,7], granting additional insights prior to developing a full-scale model. A summary of the characteristic of the bars identified during this study is presented in Table 3.

**Table 3.** Summary of minimum and maximum values observed for sandbars identified by the automated methodology in the Long Branch dataset. Bar height and slope minimums cannot fall below the 0.7 ft and 2.0%grade threshold values. All other values are derived from the sandbar toe, crest, and heel locations in the profile.

| Sandbar Characteristic Parameter | Min. Value | Max. Value |
|---|---|---|
| Bar Height (m) | 0.2 | 1.3 |
| Crest Depth (m below MHW) | 0.06 | 4.8 |
| Slope (%grade toe to crest) | 2.0 | 9.3 |
| Bar Width (m) | 14.3 | 56.3 |
| Bar Shape Parameter | 0.4 | 5.7 |

*December 2020 High-Resolution Case Study*

The high-resolution section (lines 26–70) of the December 15, 2020 survey was selected to demonstrate the automated technique as it documents a nourished beach known to have sandbars. The survey was conducted approximately three months following the completion of the nourishment in Long Branch, NJ, USA. The two-week period prior to the survey was characterized by offshore wave heights ranging from 0.3 m to 4.2 m as measured at NDBC Station 44,065 [17] (Figure 8). The hourly buoy data and the average nearshore slope of the high-resolution section were used to calculate the surf scaling parameter [7,18] (Figure 8). The surf scaling parameter is calculated as

$$\epsilon = \frac{\sigma^2 A_b}{g\, tan^2\beta}$$

where $\sigma$ is the angular wave frequency, $A_b$ is the breaking wave amplitude (assumed to be one-half the wave height at the buoy), and $\beta$ is the average intertidal beach slope at the time of the survey [18,19]. The surf scaling parameter has been used as a tool to categorize beach states [7,19] and differs from the Iribarren number, which describes the type of breaking wave [19]. The mean value of the surf scaling parameter for this period is 5.4. The one-week average value ranged from 4.6 to 6.2. These values correspond to the Wright and Short [7] beach states C and D. When $\epsilon$ is approximately 2.0, beaches typically exist in State B, and once $\epsilon > 2.5$, beaches transition to State C and D morphologies [7]. The surf scaling parameter values associated with each beach state is summarized in Table 1.

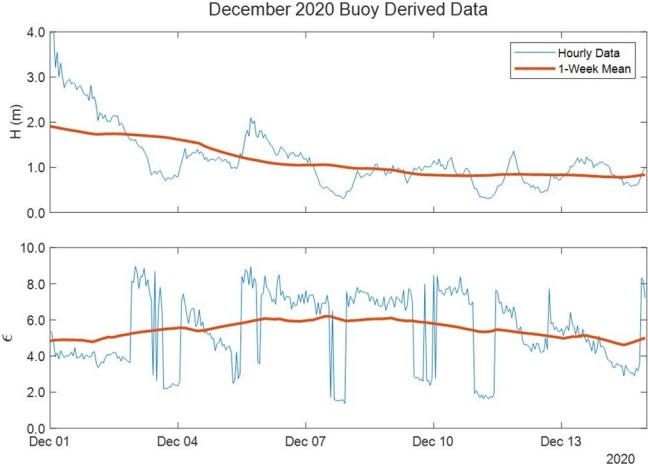

**Figure 8.** Time series data displaying wave heights (H; top) observed at NDBC Station 44,065 and the surf scaling parameter ($\epsilon$; bottom) calculated from the observed buoy data and intertidal beach slope. The blue lines represent hourly data, and the bolded red lines represent the weekly moving averages of the data.

The beach morphology observed on 15 December 2020, was characterized by well-defined sandbars, cuspate features, and an average intertidal slope of 9.5%. Sandbars were visually observed during survey activities, and the presence of bars throughout the survey area was confirmed through the generation of the DEM and extraction of profiles. These results suggest that the December beach morphology at Long Branch fits well within the State C morphological category and supports the results as determined by crest location, slopes, and trough depth identified by the automated process. Crest locations of sandbars identified as 'True' within the high-resolution area are plotted on the survey DEM (Figure 9a), and eight transects are identified to show sequentially how the bar position varies from north to south. Figure 9b displays the profiles of the selected transects with sandbar with the geomorphic features indicated; the automated toe, crest, and heel locations match well with the locations that an observer would intuitively select. Sandbars were not identified at Line 47 and Line 41 transects; however, the pattern of bars, or lack of bars, corresponds with crescentic bars and small rips. A higher resolution DEM, focused on the area around line 34 (Figure 10a), displays the bar formation in detail as well as the identified toe, crest, and heel locations (Figure 10b). This display indicates that the features identified in the automated methodology apply to larger spatial scales and that the heel locations, as determined by changepoints, apply well to depth scales and cross-shore positioning.

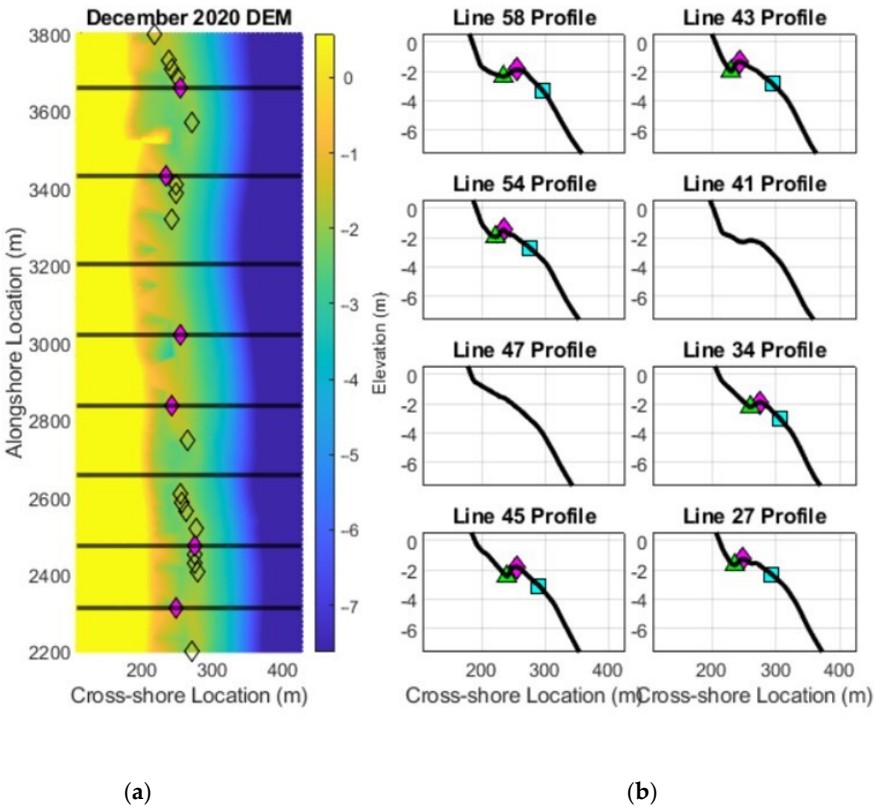

(**a**)                                                                 (**b**)

**Figure 9.** (**a**) December 2020 DEM of the high-resolution portion of the survey area with crest locations (diamond markers) and select profile transects (black lines) overlaid. From north to south, the selected transects are Line 58, Line 54, Line 47, Line 45, Line 43, Line 41, Line 34, and Line 27. Magenta diamonds indicate crest locations corresponding to the selected transects. The color bar has been chosen to highlight various morphologic features; yellow indicates areas of the beach above the MHW line, and dark blue indicates areas where the depth exceeds −7.6 m NAVD 88; orange corresponds to the sloping beach face, horns of beach cusps, and longshore sand bars; greens correspond to shallow water on either side of the bar crests, channels between bars, and cusp embayments; light blue corresponds to the transitional area on the seaward side of the sandbar where depth increases.

(**b**) Profiles displaying the submerged beach (MHW to −7.6 m NAVD88) corresponding to the select transects marked on the DEM. These transects were selected for spatial coverage of the high-resolution area. These profiles display the sand bars and present a sample of bar morphologies, including two transects in which no sandbars were observed (Lines 47 and 41). Green triangles represent the toe, magenta diamonds the crests, and cyan squares the heels of the sandbars.

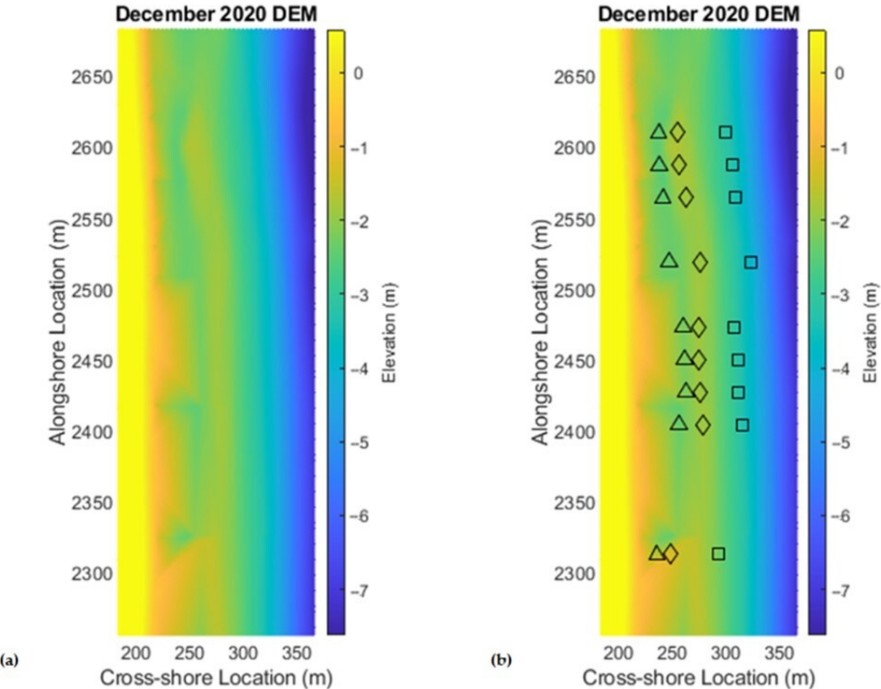

**Figure 10.** (**a**) December 2020 DEM focused on the area around Line 35 (the southernmost transect displayed in Figure 7). (**b**) December 2020 DEM with bar features overlay. Triangles indicate the toe of the bar, diamonds indicate crests, and squares indicate the bar heel. The color bar has been chosen to highlight various morphologic features; yellow indicates areas of the beach above the MHW line, and dark blue indicates areas where the depth exceeds −7.6 m NAVD 88; orange corresponds to the sloping beach face and longshore sand bars; greens correspond to shallow water on either side of the bar crests and light blue corresponds to the transitional area on the seaward side of the sandbar where depth increases.

The physical survey and automated bar identification results best fit with Wright and Short's intermediate State C: Rhythmic Bar and Beach [7] and agree with the beach state predicted by the surf scaling parameter. This classification is supported by the observed pattern of crest locations (Figure 9), trough depth, and surf scaling parameter. Although similar, State B is characterized by deeper troughs than was observed and minimal longshore variance in bar position. However, it should be noted that the overall steepness at Long Branch is greater than observed by Wright and Short [7].

## 5. Summary and Future Work

An automated technique to identify large-scale bar features was developed to rapidly assess beach survey data and identify prominent sandbars with minimum slopes of 2% grade and a height threshold of 0.2 m. This technique allows for the efficient selection of sandbar crest, toe, and heel positions without introducing subjectivity from an observer. This is achieved by defining the crest, toe, and heel in quantitative terms. The morphological characteristics can be identified on profiles from stand-alone surveys and do not require multi-year datasets to generate a reference profile. The automated technique was tested on 840 profiles collected near a recently completed beach nourishment project and successfully

identified prominent sandbars within the test data set. Furthermore, this technique is efficient and requires little processing cost to assess a large dataset.

Identification of sandbars plays a crucial role in better understanding nearshore dynamics, storm protection, beach category, and assessments of sediment storage within a littoral system. A high-resolution case study comparing the extracted features to a survey DEM supports that the methodology is appropriate and spatially sound in both the cross-shore and at-depth. The automated sandbar identification results were assessed spatially to determine beach state. The results showed agreement between the automated identification, the survey DEM, and the calculated surf scaling parameter.

Future iterations of this technique will incorporate techniques to identify other sandbar morphologies, such as multiple bars, nearshore terraces, and less prominent sandbars. Identification of terraces has been successful using reference profiles [2]; however, terraces do not have a distinct trough, and the terrace crest is not necessarily the local maximum but the point where the terrace becomes a step [2], complicating automatic extraction. Such assessments can further aid in determining beach classifications [1,7] and study sandbar morphologies and migration patterns in greater detail, including how these morphologies relate to variability in the beach face and the nearshore zone [3,8].

An interesting continuation of the current analysis would incorporate micro-level sandbar features, such as mega ripples. Such features are indicative of certain beach states [7]. The current instrumentation settings and data processing techniques require sensitivity adjustments to capture these features. More detailed surveys may provide interesting insights into understanding the evolution of various beaches and improve the determination of types of sandbars. Another option is a direct comparison of the automated results to Synthetic Aperture Radar C-Band (SAR-C) data. SAR-C is a remote sensing tool capable of penetrating the shallow depths where nearshore sandbars occur. SAR-C is typically used for global mapping and maritime ocean navigation [20]. A comparison of these methods would demonstrate the viability of the automated technique. Additionally, understanding how the survey data relates to remotely sensed data can aid in filling temporal gaps between field surveys. This is especially useful for generating a time series of profile changes or charting bar positions if limited by survey frequency.

**Author Contributions:** Conceptualization, N.Z., L.K., and J.M.; methodology, N.Z.; software, N.Z.; validation, N.Z.; formal analysis, N.Z.; investigation, N.Z., L.K., and J.M.; resources, L.K. and J.M.; data curation, N.Z.; writing—original draft preparation, N.Z.; writing—review and editing, N.Z., L.K. and J.M.; visualization, N.Z.; supervision, J.M.; project administration, N.Z., L.K., and J.M.; funding acquisition, J.M. All authors have read and agreed to the published version of the manuscript.

**Funding:** This research was funded by the New Jersey Department of Environmental Protection (NJDEP) 542 through the New Jersey Coastal Protection Technical Assistance Service (N.J.S.A. 18A:64L-1).

**Institutional Review Board Statement:** Not applicable.

**Informed Consent Statement:** Not applicable.

**Data Availability Statement:** The data that support the findings of this study are available from the corresponding author, Jon Miller, upon reasonable request.

**Acknowledgments:** Special thanks to all the Stevens CERG students and staff who aided in data collection from the inception of monitoring at Long Branch in June 2020 through to the present.

**Conflicts of Interest:** The authors declare no conflict of interest.

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
