# Peer review of "Automated Technique for Identification of Prominent Nearshore Sandbars"

_2673-964X, doi:10.3390/coasts3020009_

Round 1

Reviewer 1 Report

The manuscript entitled "Automated technique for identification of prominent nearshore sandbars" presents a novel methodology to identifying the morphological characteristics of sandbars in the nearshore. The contribution of this work to the field of coastal geomorphology is expected to be valuable. The manuscript appears to be suitable for publication after addressing comments listed below.

Comments:

Lines 57-60: Suggest presenting the morphology of prominent sandbars (A, B, C, etc.) in a table format accompanied by relevant visual tools.

Figure 2: Suggest including some bed elevations in aerial imagery.

Line 68: The extents of study area is reported in miles. suggest using km to be consistent.

Line 69: What is the elevation of the edge of the berm?

Figure 3: Suggest differentiating between the standard lines and the high resolution lines with different colors in the figure and using the term "Transect" instead of “lines”.

Line 83: The authors need to mention who collected the bathymetry data? Were the authors themself collected the data or the bathymetry data are obtained from a specific database?

Line 83: Are you referring to drone by "unmanned aerial vehicle"? If yes, better to give some specifications of the drone.

Line 86: It is important to clarify whether the cited reference [6] pertains to the methodology, equipment, or both?

Line 88: Suggest using past tense for the analysis which has been performed to get the geospatial data.

Line 91-93: Suggest re-writing this sentence and avoid using "must". It can be re-written as "All the points collected from the three methods were combined to generate a surface to analyze and capture the morphology features of the sandbars".

Line 93 through 96: Needs to be re-written. It is not clear.

Line 166:122: Might be useful to include some graphics to better explain the process.

Line 131-141: Suggest using graphical features as arrows, text, circle shapes to better explain the process.

Line 131-132: How do you come up with the 2% slope? Is it a result of some curve fitting to the data?

Line 147-149: Again, what is the basis for 0.2 m threshold for the berm height? Is it based on an average height observed in sandbars?

Figure 6: Suggest using the term "bar trough" instead of "toe". "Bar trough" is a more common terminology in the nearshore/coastal community.

Line 155: Suggest using "sandbar geometric features" instead of "parameter".

Line 158: Suggest providing definitions of the bar asymmetry and bar shape.

Line 168: Suggest using the term "observer" instead of "auditor".

Line 170-172: Suggest showing two profiles examples corresponding to before and after nourishment and show how nourishment affected the shape of the profiles and how the model can capture than.

Line 191: Suggest using "failed" instead of "rejected".

Line 197: using "and so on" is not very common in scientific documents. Also, it would be useful to cite some earlier studies for this statement. 

Line 203-205: Suggest using "nearshore processes" instead of hydrodynamics.

Line 213-222: Suggest moving this paragraph to the introduction.

Line 228-230: Suggest plotting the time series of wave heights along with the observed tidal elevations for the observation period.

Line 231: What is the "surf scaling parameter"? are you referring to the Iribarren number? It might be useful to provide a time series of this parameter along with the time series of waves and tides.

Figure 8: Supposed to have (a) and (b) labels as referred to in text.

Section 4.1: Suggest plotting time series of the bar crest and trough heights and correlate them with the time series of waves and tides to provide more context of the morphology changes and how the model can capture that.

Author Response

Thank you for your thoughtful comments and taking the time to clearly state any concerns and suggestions.

Comments:

Lines 57-60: Suggest presenting the morphology of prominent sandbars (A, B, C, etc.) in a table format accompanied by relevant visual tools.

  • A table has been added summarizing the various beach state features and association with prominent sandbars.

Figure 2: Suggest including some bed elevations in aerial imagery.

  • Due to space constraints, an additional panel was added to Figure 3 to show elevation rather than including the addition in Figure 2.

Line 68: The extents of study area is reported in miles. suggest using km to be consistent.

  • Good catch! Thanks!

Line 69: What is the elevation of the edge of the berm?

  • This is addressed in the updated text and in the updated Figure 3 caption.

Figure 3: Suggest differentiating between the standard lines and the high resolution lines with different colors in the figure and using the term "Transect" instead of “lines”.

  • The figure was updated, and the term “transect” was also updated within the text.

Line 83: The authors need to mention who collected the bathymetry data? Were the authors themself collected the data or the bathymetry data are obtained from a specific database?

  • This has been clarified in the text.

Line 83: Are you referring to drone by "unmanned aerial vehicle"? If yes, better to give some specifications of the drone.

  • This has been clarified and specifications of the UAV/drone have been included.

Line 86: It is important to clarify whether the cited reference [6] pertains to the methodology, equipment, or both?

  • This has been clarified within the text.

Line 88: Suggest using past tense for the analysis which has been performed to get the geospatial data.

  • Thank you; the rest of the section has also been reviewed for verb tense.

Line 91-93: Suggest re-writing this sentence and avoid using "must". It can be re-written as "All the points collected from the three methods were combined to generate a surface to analyze and capture the morphology features of the sandbars".

 Line 93 through 96: Needs to be re-written. It is not clear.

  • Comments Line 91-93 and Line 93 through 96 here together. The section has been rewritten for clarity.

Line 166:122: Might be useful to include some graphics to better explain the process.

Line 131-141: Suggest using graphical features as arrows, text, circle shapes to better explain the process.

  • Comments Line 166:122 and Line 131-141 are addressed here together. A flow chart has been added to the manuscript to help explain the decision process.
  • Per Reviewer 3’s comments, original figures 4, 5, and 6 have also been updated and combined into a single figure for clarity.

Line 131-132: How do you come up with the 2% slope? Is it a result of some curve fitting to the data?

  • An explanation was added to the text to address the reasoning and functionality of the 2% threshold value.

Line 147-149: Again, what is the basis for 0.2 m threshold for the berm height? Is it based on an average height observed in sandbars?

  • This was addressed in the original version in the introduction and has been reiterated in this section for clarity.

Figure 6: Suggest using the term "bar trough" instead of "toe". "Bar trough" is a more common terminology in the nearshore/coastal community.

  • The figure and caption have been updated accordingly.

Line 155: Suggest using "sandbar geometric features" instead of "parameter".

  • Suggested change made.

Line 158: Suggest providing definitions of the bar asymmetry and bar shape.

  • The text has been clarified and the bar shape parameter is defined. A figure was also added to better explain the features used to calculate the bar shape parameter.

Line 168: Suggest using the term "observer" instead of "auditor".

  • All instances of “auditor” have been replaced with “observer”.

Line 170-172: Suggest showing two profiles examples corresponding to before and after nourishment and show how nourishment affected the shape of the profiles and how the model can capture than.

  • A figure has been added displaying three profiles (pre-nourishment, 1 week post nourishment, and 3 months post nourishment). This figure shows the changes in the cross-shore that limit applicability of reference profiles. Our model is not limited by these cross-shore changes.

Line 191: Suggest using "failed" instead of "rejected".

  • Suggested change made.

Line 197: using "and so on" is not very common in scientific documents. Also, it would be useful to cite some earlier studies for this statement. 

  • The sentence has been updated.

Line 203-205: Suggest using "nearshore processes" instead of hydrodynamics.

  • Suggested change made.

Line 213-222: Suggest moving this paragraph to the introduction.

  • This paragraph was moved to the end of the introduction section.

Line 228-230: Suggest plotting the time series of wave heights along with the observed tidal elevations for the observation period.

Line 231: What is the "surf scaling parameter"? are you referring to the Iribarren number? It might be useful to provide a time series of this parameter along with the time series of waves and tides.

  • Comments Line 228-230 and Line 231 are addressed here together. A figure was created displaying observed wave heights and the surf scaling parameter. NOAA does not have observed tidal data at the study site for the study period.
  • An explanation of the surf scaling parameter was added to the text, as well as information regarding its calculation. The difference between the surf scaling parameter and Iribarren number (or surf similarity parameter) is also addressed.

Section 4.1: Suggest plotting time series of the bar crest and trough heights and correlate them with the time series of waves and tides to provide more context of the morphology changes and how the model can capture that.

  • Thank you for this recommendation; it will certainly be applied to future work to assess changes between monthly surveys.

Figure 8: Supposed to have (a) and (b) labels as referred to in text.

  • The labels have been corrected and the figure caption has been updated accordingly.

Reviewer 2 Report

1. The paper focuses specifically on the study of the automated technique for identification of prominent nearshore sandbars.

2. The authors should also present what novelty they bring to the current state of knowledge and the aim and objectives should be added in the section of the 1. Introduction

3. The abstract needs to be redone, whereas, the general information may be included in this section.

4. The prominent nearshore sandbars map should be added to the ground truth validation of the photographic documents and the area wise change detection sand bar map should be added in the section of the 4.0 Results & Discussion.

5. All figures are not properly anoted. The tables, figures are not lucidly explained in 'Result and Discussion' but few things can be incorporated to justify the relation which can give a better result like high resolution SAR-C band data should be added  for the assessing of the spatio-temporal ‘changing point of sand fan lobes deposits’ in present study area.

6. Micro level rhythmic bar and beach features should be incorporate in the section of the 4.0 Results & Discussion

7. Recommendations should be added to the manuscripts for better understanding by the researchers.

Author Response

  1. The paper focuses specifically on the study of the automated technique for identification of prominent nearshore sandbars.
  • Thank you for taking the time to review and comment.

  1. The authors should also present what novelty they bring to the current state of knowledge and the aim and objectives should be added in the section of the 1. Introduction
  • Verbiage was added to the first paragraph to reiterate the objective and novelty of our automated sandbar assessment technique. A paragraph was also added to the end of the introduction section expanding on the applications that this technique may be suited for.

  1. The abstract needs to be redone, whereas, the general information may be included in this section.
  • The abstract has been reworded to be more specific about the aim of research.

  1. The prominent nearshore sandbars map should be added to the ground truth validation of the photographic documents and the area wise change detection sand bar map should be added in the section of the 4.0 Results & Discussion.
  • These concerns have been addressed in accordance with specific suggestions from other reviewers. For example, a new figure (Figure 7) has been added to show the cross-shore variation at three timesteps related to the beach nourishment. Figure 9 (formerly 7) includes additional transects on the DEM with associated profiles to show the bar selection on ground truth survey data. Additionally, Section 4.0 was updated in accordance with your Comment 6. and Section 4.1 has undergone significant revision.

  1. All figures are not properly anoted. The tables, figures are not lucidly explained in 'Result and Discussion' but few things can be incorporated to justify the relation which can give a better result like high resolution SAR-C band data should be added  for the assessing of the spatio-temporal ‘changing point of sand fan lobes deposits’ in present study area.
  • The incorrect annotations were corrected.
  • Thank you for the recommendation of including SAR-C data. This is beyond the scope of the current paper but would be interesting to include for a future publication. This is also addressed in your comment 7 regarding recommendations.
  • Several figures were added throughout the manuscript and captions were modified to be more descriptive.

  1. Micro level rhythmic bar and beach features should be incorporate in the section of the 4.0 Results & Discussion
  • A paragraph has been added (2nd paragraph of the section) to address this concern and to clarify that the overall intent is to identify large scale bar features.
  • This is also further addressed in continuing work recommendations.

  1. Recommendations should be added to the manuscripts for better understanding by the researchers.
  • The summary and future work section has been rewritten to more clearly state continuing work and provide recommendations for future studies. These recommendations include comparing the automated sandbar identifications with SAR-C data and future studies assessing micro-level sandbar features.

Additional references have also been added.

Reviewer 3 Report

The paper presents an automated technique for identifying the existence of the prominent nearshroe sandbars. The topic is quite interesting and the efficiency of the method  is well documented. I thus recommend the acceptance of the paper after a minor revision.

1.  Figs. 4-6 should be integrated into one.

2. For a real applications, for example the case in 4.1, I would like to see the continuous variation of bar positions (crest, trough and heel) in Figs.7-8. This could be done by specifying more transects. That would be very usesful for analyzing the sandbar pattern in longshore directioin.

Author Response

The paper presents an automated technique for identifying the existence of the prominent nearshore sandbars. The topic is quite interesting and the efficiency of the method is well documented. I thus recommend the acceptance of the paper after a minor revision.

  • Thank you!

Comments:

  1. Figs. 4-6 should be integrated into one.
  • The figures have been combined into one with an A-B-C designation.
  1. For a real applications, for example the case in 4.1, I would like to see the continuous variation of bar positions (crest, trough and heel) in Figs.7-8. This could be done by specifying more transects. That would be very useful for analyzing the sandbar pattern in longshore direction.
  • Additional transects were added to Figure 9 (formerly Figure 7) to show varying crest positions north to south, as well as examples where no bar was identified. The additional transect profiles show sequentially how the bar moves.

Round 2

Reviewer 1 Report

All my comments have been addressed and the manuscript is ready for publication.